Glycosylceramide modifies the flavor and metabolic characteristics of sake yeast

Ferdouse Jannatul 1
Yamamoto Yuki 2
Taguchi Seiga 2
Yoshizaki Yumiko 1
Takamine Kazunori 1
Kitagaki Hiroshi 1 2 ktgkhrs@cc.saga-u.ac.jp
1 Department of Biochemistry and Applied Biosciences, United Graduate School of Agricultural Sciences, Kagoshima University , Kagoshima , Japan
2 Department of Environmental Science, Faculty of Agriculture, Saga University , Saga , Japan
Portillo Maria del Carmen
Electronic publication date: 2018 May 10
Publication date: 2018
Volume: 6
Electronic Location ID: e4768
Received 2018 Jan 20; Accepted 2018 Apr 24
Copyright: © 2018 Ferdouse et al.
Copyright year: 2018
Copyright holder: Ferdouse et al.
License: This is an open access article distributed under the terms of the Creative Commons Attribution License, which permits unrestricted use, distribution, reproduction and adaptation in any medium and for any purpose provided that it is properly attributed. For attribution, the original author(s), title, publication source (PeerJ) and either DOI or URL of the article must be cited.
License URL: https://creativecommons.org/licenses/by/4.0/

Keywords: Aspergillus oryzae, Koji, Glycosylceramide, Saccharomyces cerevisiae, Sphingolipid, Sake yeast, Volatile ester, Flavor

Funding: JSPS KAKENHI 15K07363 This study was financially supported by JSPS KAKENHI 15K07363 to Hiroshi Kitagaki. There was no additional external funding received for this study. The funders had no role in study design, data collection and analysis, decision to publish, or preparation of the manuscript.

==============================
In the manufacture of sake, Japanese traditional rice wine, sake yeast is fermented with koji, which is steamed rice fermented with the non-pathogenic fungus Aspergillus oryzae. During fermentation, sake yeast requires lipids, such as unsaturated fatty acids and sterols, in addition to substances provided by koji enzymes for fermentation. However, the role of sphingolipids on the brewing characteristics of sake yeast has not been studied. In this study, we revealed that glycosylceramide, one of the sphingolipids abundant in koji, affects yeast fermentation. The addition of soy, A. oryzae, and Grifola frondosa glycosylceramide conferred a similar effect on the flavor profiles of sake yeast. In particular, the addition of A. oryzae and G. frondosa glycosylceramide were very similar in terms of the decreases in ethyl caprylate and ethyl 9-decenoate. The addition of soy glycosylceramide induced metabolic changes to sake yeast such as a decrease in glucose, increases in ethanol and glycerol and changes in several amino acids and organic acids concentrations. Tricarboxylic acid (TCA) cycle, pyruvate metabolism, starch and sucrose metabolism, and glycerolipid metabolism were overrepresented in the cultures incubated with sake yeast and soy glycosylceramide. This is the first study of the effect of glycosylceramide on the flavor and metabolic profile of sake yeast.

Introduction

Japanese traditional rice wine, sake, uses rice as its raw material. Therefore, rice starch is used as a source of carbohydrates, and saccharification of starch is necessary for subsequent fermentation by sake yeast Saccharomyces cerevisiae. As a saccharifier of starch, koji, which is steamed rice fermented with the non-pathogenic fungus Aspergillus oryzae, has been used traditionally in Japan, as a Japanese version of the malt used in beer in western countries. Koji is then mixed with steamed rice and sake yeast, and thus simultaneous saccharification and fermentation occurs (Kitagaki & Kitamoto, 2013).

Koji contains various enzymes, including glycosidases, proteases, and lipases. The role of these koji enzymes on sake brewing has been studied intensively. Indeed, yeast can synthesize most substances, including saturated fatty acids, from substances provided from rice and degraded by koji enzymes, allowing them to proliferate and ferment. However, in addition to these substances, yeast needs lipids other than saturated fatty acids. Past studies have been limited to unsaturated fatty acids and β-sitosterol. For example, unsaturated fatty acids have been shown to facilitate fermentation and decrease isoamyl acetate production (Ohta & Hayashida, 1983; Fujii et al., 1997; Mason & Dufour, 2000). Unsaturated fatty acids, β-sitosterol, and phospholipids synergistically affect the fusel alcohols content of beer and decrease the content of volatile esters and medium chain-length fatty acids (Taylor, Thurston & Kirsop, 1979). In addition, grape phytosterol increases the fermentation rate of yeast (Luparia et al., 2004). However, there has been no report on the effect of sphingolipids on yeast fermentation.

Sphingolipids are a class of lipids that contain amide bonds of sphingoid bases and fatty acids. Serine and palmitoyl-CoA are conjugated to form sphinganine, which is reduced to sphingoid bases, such as phytosphingosine, sphingosine, and dihydrosphingosine. Fatty acids are amide-linked to the sphingoid bases to form ceramides. In ceramides, either phosphate or carbohydrates are linked to the hydroxyl bond at the first position. Phosphate is ester-linked to the first hydroxyl bond of ceramides, and further diester-linked to inositol or choline to form acid complex sphingolipids, such as inositol-phosphoceramide (IPC), mannose-inositol-phosphoceramide (MIPC), and mannose-(inositol phosphate)2-ceramide (M(IP)2C) (Dickson et al., 1997) or sphingomyelin (Okazaki, Bell & Hannun, 1989). Carbohydrates, including glucose, galactose, sialic acid, and/or their conjugates, are acetal-linked to the first hydroxyl bond of ceramide to form glycosylceramides. Exceptionally, S. cerevisiae lacks a glycosylceramide-synthesizing enzyme (Saito et al., 2006). Sphingolipids form rafts in the lipid bilayer (Simons & Ikonen, 1997) and also function as signaling lipids (Hannun & Obeid, 2008).

In previous studies, we revealed that one of the sphingolipids, glycosylceramide, having galactose (19.2%) or glucose (80.8%) as the monohexosyl moiety, 9-methyl-4,8-sphingadienine as the sphingoid base moiety, and 2′-hydroxyoctadecanoic acid as the fatty acid moiety, is contained in shochu and sake koji (Hamajima et al., 2016; Hirata et al., 2012). Furthermore, we have revealed that the glycosylceramide contained in shochu koji increased the alkali tolerance and modified the flavor profile of shochu yeast (Sawada et al., 2015). However, the effect of koji glycosylceramide on the fermentation profile of sake yeast remains unknown. Therefore, in the present study, we investigated the effect of koji glycosylceramide on the brewing characteristics of sake yeast.

Materials and Methods

Strains, reagents, and materials

The S. cerevisiae sake yeast (isolated and distributed in 2003 from the Brewing Society of Japan, K7-4) was obtained from the Brewing Society of Japan (Tokyo, Japan). Conidia of A. oryzae were obtained from Higuchi Moyashi Co., Ltd. (Osaka, Japan). Analytical grade reagents were used. Glycosylceramides of soy and Grifola frondosa were purchased from Funakoshi Co. Ltd (Tokyo, Japan), and that of A. oryzae was purified from the mycelia of A. oryzae.

Culture of A. oryzae

Conidia of A. oryzae (10 mg) were dissolved in 1 ml of sterile water, and 100 μl of the solution was inoculated onto potato dextrose agar (0.4% potato starch, 2% dextrose, 2% agar) and incubated at 30 °C for 3 days. An aliquot of the culture was inoculated into 500 ml of potato dextrose medium (0.4% potato starch, 2% dextrose) and incubated at 30 °C and 200 rpm for five days. Mycelia of A. oryzae were washed with sterile water three times, freeze-dried for two days, and ground with a mortar.

Lipid extraction

Dried A. oryzae mycelia (0.12 g) in a glass tube with a screw cap were added with 2 ml of chloroform/methanol (1:1 v/v) and vortexed for 1 min. To the solution, 2 ml of 0.8M KOH dissolved in methanol was added and incubated at 42 °C and 160 rpm for 30 min. Subsequently, 5 ml of chloroform and 2.25 ml of water was vortexed until saponification, and centrifuged at 700 × g for 10 min. The lower organic layer was recovered and evaporated under a vacuum.

Separation and detection of lipids using thin layer chromatography

Separation and detection of lipids were performed basically as previously described (Takahashi et al., 2014). The extracted lipids (500 μl) were dried, dissolved in 50 μl of chloroform/methanol (2:1 v/v), and 20 μl of the solution was spotted onto a thin layer chromatography (TLC) plate (Silica gel 60 plate, Merck Millipore Inc., Darmstadt, Germany). The dried plate was developed with saturated chloroform:methanol:acetic acid:water (20:3.5:2.3:0.7 v/v). Cerebroside (Matreya Inc., Pleasant Gap, PA, USA) was used as an internal standard. To detect glycosylceramide, 2 mg/ml of orcinol in 70% sulfuric acid was sprayed onto the TLC plate and heated at 100 °C for 40 min.

Semi-purification of glycosylceramide using column chromatography

Crude extracts of sphingolipids were evaporated under a vacuum, solubilized in 3 ml of chloroform, and applied to silica gel chromatography (diameter 30 mm, column length 300 mm, Silica Gel 60 70–230 mesh, Nacalai Tesque Inc., Kyoto, Japan). First, 600 ml of chloroform was used as an eluent to remove contaminants. Then, 10 ml of ethyl acetate:methanol (v/v, 9:1) was added 24 times. An aliquot (800 μl) of each 10 ml fraction was dried, dissolved in 50 μl of chloroform/methanol (2:1 v/v), and was spotted onto a TLC plate as described above. An Rf value of 0.4 was adopted as the glycosylceramide fraction, as previously reported (Takakuwa et al., 2005).

Purification of glycosylceramide using high performance liquid chromatography (HPLC)

Fractions that contained glycosylceramides were evaporated under a vacuum, dissolved in 1.5 ml of chloroform, and injected into a 0.5 ml injection loop. Purification was performed as follows; particle size: 5 μm, diameter: 4.6 mm, length: 250 mm (Inertsil SIL 100A, GL Science Inc., Tokyo, Japan), column temperature; 30 °C, buffer A: chloroform, buffer B: 95% methanol, gradient: 0.01 min A 100%, 60 min A 85%, 80 min A 10%, 90 min A 10%, 90.01 min A 100%, 100 min A 100%, 100.01 min STOP; rate: 0.7 ml/min. Eluents at 40–50 min were sampled at 1 min intervals, and aliquots of the fractions were dried and applied to TLC analysis. The fractions that were confirmed to contain purified glycosylceramide were dried under a vacuum (Fig. S1) and dissolved in ethanol (4 μg/μl).

Co-culture of sake yeast and glycosylceramide

Sake yeast was incubated in 3 ml of YPD medium (1% yeast extract, 2% bactopeptone, 2% glucose) at 30 °C for 24 h. Cells were recovered and washed with sterile water. Koji extract solution (5 ml, pregelatinized koji weighing 5 g was mixed with water and incubated at 55 °C overnight, and the liquid fraction was recovered) was inoculated with nonidet P-40 (final 0.0015% v/v), with or without 50 μl of 4 μg/μl glycosylceramide (either from A. oryzae, soy, or G. frondosa) dissolved in ethanol and 106 cells/ml sake yeast (the final glycosylceramide concentration was 40 μg/ml). Sake yeast is exposed to 40 μg/ml koji glycosylceramide during fermentation; therefore, this concentration was adopted. Sterile liquid paraffin (1 ml) was overlaid onto the medium, and the samples were incubated at 15 °C for one week. The volatile and non-volatile compounds of the culture were analyzed. Flow chart of the experimental procedures is shown in Fig. 1.

Figure 1 Flow chart of the experimental procedure.

Sample collection for metabolites analysis

After one week of fermentation with or without glycosylceramide, the fermented cultures were centrifuged at −9 °C and 3,200 × g for 3 min. The supernatant (5 ml) was divided into two plastic tubes (2.5 ml each) and stored at 4 °C for volatile compound analysis or at −27 °C for non-volatile compounds analysis. The pellet was resuspended in 3 ml of MilliQ water (Millipore Inc., Darmstadt, Germany) and centrifuged again under the same conditions. The pellet was soaked in liquid nitrogen for 3.5 min, freeze-dried for 12 h, and stored at −80 °C.

Sample preparation for the analysis of volatile compounds

Fermented cultures were centrifuged at 4 °C and 3,200 × g for 3 min, and the supernatant was collected. For target products analysis, 900 μl of fermentation broth and 100 μl of internal standard mixture (methyl hexanoate at 5 mg/l and n-amyl alcohol at 200 mg/l) were placed into a 10-ml glass vial on ice. The vial containing the fermentation broth and an internal standard mixture was sealed with a magnetic cap and subjected to GC/MS analysis, as previously described (Yoshizaki et al., 2010).

Gas chromatography mass spectrometry analysis of volatile compounds

Aroma compounds of the fermented cultures were analyzed using headspace gas chromatography mass spectrometry (GC/MS) (GC-2010, GCMS-QP2010; Shimadzu, Kyoto, Japan) equipped with a DB-WAX column (60 m; internal diameter, 0.25 mm; 0.5 m; Agilent Technologies, Palo Alto, CA, USA) as previously described (Yoshizaki et al., 2010). Experiments were performed in triplicate from respective independent cultures.

Sample preparation for extracellular non-volatile compounds

Sample preparation for extracellular non-volatile compounds was performed as described previously (Kadowaki et al., 2017) with minor modifications. The fermented cultures (100 μl) or freeze-dried cells (10 mg) were mixed with 1 ml of chloroform:methanol:water (2:5:2) for extraction. Ribitol solution (0.2 mg/ml, 60 μl) was added to the solution and incubated for 30 min at 30 °C with shaking at 1,500 rpm. The supernatant (900 μl) was collected by centrifugation (4 °C 16,000 × g for 3 min) and 400 μl of MilliQ water (Millipore Inc., Darmstadt, Germany) was added, mixed, and centrifuged again under the same conditions. The supernatant (800 μl) was evaporated for 3 h and freeze-dried for 12 h. Methoxyamine (20 mg/ml dissolved in pyridine, 100 μl) was mixed with the freeze-dried extract and incubated at 30 °C for 90 min with shaking at 1,500 rpm. N-methyl-N-(trimethylsilyl) trifluoroacetamide (MSTFA) (50 μl) was added and again incubated at 37 °C for 30 min with shaking at 1,500 rpm. The solution (70 μl) was transferred to a vial and subjected to gas chromatography flame ionization detector (GC/FID) analysis.

Gas chromatography flame ionization detector analysis of non-volatile compounds

Non-volatile metabolic compounds, which are produced during fermentation with or without glycosylceramide, were analyzed using a GC/FID (GC-2014, Gas Chromatograph, Shimadzu, Kyoto, Japan) with a CP Sil8CB column (30 m × 0.25 mm × 0.25 μm; Agilent Technologies, Palo Alto, CA, USA). The carrier gas was nitrogen, with a column headspace pressure of 73.9 kPa and a flow rate of 0.97 ml/min. The gas chromatography temperature program was as follows: 60 °C for 2 min, increased to 320 °C at 13 °C /min, and held for 17 min. The split ratios for extracellular metabolites were 10 and two, respectively. The data were analyzed using GC/FID solution software (Labsolution, Shimadzu, Kyoto, Japan). All metabolite concentrations were normalized using ribitol as an internal control. Experiments were performed from respective independent seven (control) or eight (soy glycosylceramide-added) cultures.

Measurement of ethanol concentration

The ethanol concentrations of the fermented cultures were analyzed using a contact combustion system with an alcohol densitometer (Alcohol Checker YSA-200; Yazaki Meter Co. Ltd., Tokyo, Japan) according to the manufacturer’s instructions, and as described previously (Katou et al., 2008).

Statistical analysis

The statistical significances of differences among the averages of volatile flavors were judged using Student’s t-test and the false discovery rate. The experimental results were expressed as means ± standard error of the means. For heatmap analysis and integrated pathway analysis, MetaboAnalyst (Xia et al., 2015) was used. Heatmaps were based on interquantile range data filtering, normalization by sum, clustering with Ward’s method, distance measure with Euclidean, standardization by autoscale features and without data scaling.

Results

First, the flavor compounds of the cultures fermented with sake yeast added with or without glycosylceramides were analyzed (Tables S1–S6). To obtain information on the specificity of the effect of the chemical structures of glycosylceramide, glycosylceramides from A. oryzae, soy, and G. frondosa were used. A concentration of 40 μg/ml was adopted, because koji contains approximately 240 μg/g of glycosylceramide (Sawada et al., 2015) and koji is contained at a ratio of one to six of water and rice in the first step of sake brewing. As a result, addition of A. oryzae, soy and G. frondosa glycosylceramides showed patterns distinct from vehicle control (ethanol-added culture) (Fig. 2). In particular, A. oryzae and G. frondosa glycosylceramides were clustered in proximity, showing similar decreases in ethyl caprylate and ethyl 9-decenoate and increases in 2-phenylethyl acetate and phenylethyl alcohol (Table 1). Soy glycosylceramide had a similar effect to A. oryzae in terms of the increase in 2-phenylethyl acetate and decreases in ethyl 9-decenoate, isoamylalcohol, and acetoin (Table 2).

Figure 2 Heatmap of volatile compounds in the culture of sake yeast incubated with or without glycosylceramide.

Sake yeasts were incubated in synthetic medium containing 40 μg/ml A. oryzae (A1 and A2), soy (S), G. frondosa (G) glycosylceramide or their vehicle control ethanol (E1 and E2) and nonidet P-40 (final concentration 0.0015% v/v) at 15 °C for one week. Volatile compounds were analyzed using headspace gas chromatography mass spectrometry (GC/MS). A heatmap of volatile compounds was created with Metaboanalyst.

Table 1 Volatile compounds in the culture of sake yeast added with or without glycosylceramide of A. oryzae and G. frondosa.

Volatile compounds	Control	GlcCer of A. oryzae	GlcCer of G. frondosa	
Relative percentage (%)	Relative percentage (%)	p value	Relative percentage (%)	p value	
Ethyl caprylate	100.0 ± 13.1	56.4 ± 1.78	0.0272*	50.4 ± 1.07	0.0185*	
Ethyl 9-decenoate	100.0 ± 20.9	38.4 ± 5.49	0.0401*	26.2 ± 2.35	0.0227*	
2-Phenylethyl acetate	100.0 ± 19.6	351 ± 34.6	0.00338*	431 ± 27.7	0.000671*	
Phenylethyl alcohol	100 ± 3.59	159 ± 19.3	0.0342*	216 ± 9.56	0.000376*	
Notes:

Sake yeasts were incubated in synthetic medium with or without 40 μg/ml glycosylceramide of A. oryzae and G. frondosa and nonidet P-40 (final 0.0015% v/v) at 15 °C for one week. Volatile compounds were analyzed using headspace gas chromatography mass spectrometry (GC/MS). GlcCer indicates glycosylceramide. The results are the mean values with standard errors of triplicate independent experiments. The relative percentage as compared to the mean value of the control culture is shown. Volatile compounds which were significantly different between control and GlcCer-added culture as judged by false discovery rate (p < 0.05, indicated by *) are described. P values were calculated using unpaired one-tailed Student’s t-test. Further details are described in “Materials and Methods.”

Table 2 Volatile compounds in the culture of sake yeast added with or without glycosylceramide of A. oryzae and soy.

Volatile compounds	Control	GlcCer of A. oryzae	GlcCer of soy	
Relative percentage (%)	Relative percentage (%)	p value	Relative percentage (%)	p value	
Isoamyl alcohol	100 ± 4.16	75.1 ± 5.18	0.0189*	97.6 ± 1.93	0.347	
Acetoin	100 ± 7.51	44.7 ± 8.22	0.00772*	84.7 ± 8.58	0.168	
Ethyl 9-decenoate	100 ± 13.4	35.2 ± 2.12	0.00886*	61.5 ± 3.23	0.0427*	
2-Phenylethyl acetate	100 ± 12.9	125 ± 24.3	0.249	192 ± 18.9	0.0150*	
Phenylethyl alcohol	100 ± 5.00	90.2 ± 14.0	0.310	143 ± 10.6	0.0200*	
Notes:

Sake yeasts were incubated in synthetic medium with or without 40 μg/ml glycosylceramide of A. oryzae and soy and nonidet P-40 (final 0.0015% v/v) at 15 °C for one week. Volatile compounds were analyzed using headspace gas chromatography mass spectrometry (GC/MS). GlcCer indicates glycosylceramide. The results are the mean values with standard errors of triplicate independent experiments. The relative percentage as compared to the mean value of the control culture is shown. Volatile compounds which were significantly different between control and GlcCer-added culture as judged by false discovery rate (p < 0.05, indicated by *) are described. P values were calculated using unpaired one-tailed Student’s t-test. Further details are described in “Materials and Methods.”

Next, since all glycosylceramides showed overall similar trends, the metabolites in the cultures fermented with sake yeast added with soy glycosylceramide were analyzed (Fig. 3; Table 3). As a result, independent cultures incubated with or without soy glycosylceramide were divided into respective distinct clusters (control culture and soy glycosylceramide-added culture), except CONT_6_007, SOY_1_004, and SOY_3_003 (Fig. 3). Many organic acids and amino acids, such as succinate/glycine, malate, glutamate, valine, methionine, pyruvate and threonine, and glycerol were increased and leucine was decreased in sake yeasts added with soy glycosylceramide. (Table 3). The decrease in glucose (Table 3) and increase in ethanol (Fig. 4) indicated an increased fermentation ability of sake yeast incubated with glycosylceramide, which was consistent with a previous study of shochu yeast (Sawada et al., 2015).

Figure 3 Heatmap of extracellular metabolite concentrations in the culture of yeast incubated with or without soy glycosylceramide.

Sake yeasts were incubated in synthetic medium containing 40 μg/ml soy glycosylceramide and nonidet P-40 (final concentration 0.0015% v/v) at 15 °C for one week. Metabolites of the cultures were derivatized with methoxyamine and MSTFA, analyzed using GC-FID and normalized using ribitol. A heatmap of metabolites was created with Metaboanalyst.

Table 3 Metabolite concentrations in the culture of sake yeast added with or without soy glycosylceramide.

Metabolic compounds	Control	GlcCer of soy	
Relative percentage (%)	Relative percentage (%)	p value	
Glycerol	100 ± 2.55	110 ± 1.14	0.00140	
Succinate/glycine	100 ± 4.05	116 ± 2.68	0.00376	
Malate	100 ± 2.54	116 ± 3.55	0.00191	
Glucose	100 ± 5.17	71.7 ± 3.05	0.000513	
Glutamate	100 ± 0.501	101 ± 0.315	0.0392	
Leucine	100 ± 2.73	89.8 ± 1.86	0.00770	
Valine	100 ± 5.12	128 ± 4.62	0.00225	
Methionine	100 ± 7.23	127 ± 6.12	0.0130	
Pyruvate	100 ± 15.5	222 ± 28.3	0.00235	
Threonine	100 ± 13.8	146 ± 9.62	0.0125	
Notes:

Sake yeasts were incubated in synthetic medium with or without 40 μg/ml soy glycosylceramide and nonidet P-40 (final 0.0015% v/v) at 15 °C for one week. Metabolites derivatized with methoxyamine and MSTFA, analyzed using gas chromatography flame ionization detector (GC/FID) and normalized using ribitol. GlcCer indicates glycosylceramide. The results are the mean values with standard errors of seven (control) or eight (GlcCer of soy) independent experiments. The relative percentage as compared to the mean value of the control culture is shown. P values were calculated using unpaired one-tailed Student’s t-test. Further details are described in “Materials and Methods.”

Figure 4 Ethanol concentrations (%(vol/vol)) of the culture of sake yeast added with or without soy glycosylceramide.

Sake yeasts were incubated in synthetic medium with or without 40 μg/ml soy glycosylceramide and nonidet P-40 (final 0.0015% v/v) at 15 °C for one week. The ethanol concentration of fermented culture was analyzed using a contact combustion system with an alcohol densitometer. The results are the mean values with standard errors of triplicate independent experiments. The statistical significance of the difference between the averages was assessed using the unpaired one-tailed Student’s t-test (***, p < 0.001).

To extract information about the metabolism occurring in the yeast cells incubated with glycosylceramide, the metabolome data of the medium incubated with or without glycosylceramide were analyzed using MetaboAnalyst (Xia et al., 2015). Metabolites (glycerol, succinate/glycine, malic acid, glucose, leucine, valine, glutamate, methionine, pyruvate, and threonine), which were significantly different (p < 0.05) between medium incubated with or without soy glycosylceramide, were selected and further analyzed. As a result, several metabolisms, such as pyruvate metabolism, the TCA cycle, starch and sucrose metabolism, and glycerolipid metabolism were significantly overrepresented in the medium incubated with glycosylceramide (Fig. 5). This result indicated that extracellularly added glycosylceramide has certain effects on the metabolic pathways described above, although the mechanism remains to be determined.

Figure 5 Pathway analysis of extracellular metabolites of sake yeast incubated with or without soy glycosylceramide.

The normalized values of metabolites (glycerol, succinate/glycine, malic acid, glucose, leucine, glutamate, valine, methionine, pyruvate, and threonine), which were significantly different (p < 0.05) between medium incubated with and without soy glycosylceramide, were used as independent variables for Metaboanalyst.

Discussion

In this study, we first revealed the effect of glycosylceramides on the metabolic and flavor profiles of sake yeast. Soy, A. oryzae, and G. frondosa glycosylceramides conferred similar effects on the flavor profiles of sake yeast; addition of A. oryzae and G. frondosa glycosylceramides showed especially high similarity, providing a specificity of the 9-methyl base of the sphingoid base. Addition of soy glycosylceramide induced significant changes to the concentrations of glucose, ethanol, glycerol, and several amino acids and organic acids. Several metabolic pathways were altered in sake yeast incubated with soy glycosylceramide. Considering that the content of glycosylceramide differs among koji samples (Sawada et al., 2015), these results suggested that koji exerts its effects on yeast fermentation characteristics through glycosylceramide, and that the effect of koji on sake brewing should be evaluated in terms of the effect of the quantity of glycosylceramide contained in koji.

Alcohol fermentation is performed without oxygen, which is required for synthesis of specific lipids; therefore, it has been documented that unsaturated fatty acids (Fujii et al., 1997; Fujiwara et al., 1998) and sterols (Belviso et al., 2004), which involve molecular oxygen, are required for progression of yeast fermentation. In addition, yeast incubated with or without oxygen (Ishtar Snoek & Yde Steensma, 2007) or that incubated after diauxic shift (Kitagaki et al., 2009) have different gene expression profiles or yeast incubated without oxygen change their cell wall proteins (Kitagaki, Shimoi & Itoh, 1997). However, the role of sphingolipids in yeast during fermentation has not been described until our recent study (Sawada et al., 2015).

Sphingolipids have several characteristics that are different from other lipids. For example, they are amide-linked and have long fatty acid lengths (C20–26) (Kitagaki et al., 2007). The mechanism of glycosylceramide’s effect on yeast fermentation characteristics seems to be through its increasing effect on the yeast membrane fluidity. Indeed, we have shown in a previous report that glycosylceramide shortens the average fluorescence lifetime of τ of shochu yeast, as measured by trimethylamine–diphenylhexatriene (Sawada et al., 2015). Consistent with this hypothesis, we found that volatile esters were decreased in sake yeast incubated with glycosylceramide (Fig. 2; Tables 1 and 2).

The effect of glycosylceramide on the physiology of yeast elucidated in this study suggests its impact on the membrane characteristics. It was reported that the addition of unsaturated fatty acid, which also increases membrane fluidity, decreases ethyl ester flavors (Moonjai et al., 2003; Saerens et al., 2008). It was also reported that a 9-methyl base is essential for membrane fluidity (Singh et al., 2012; Takakuwa et al., 2002). However, the specificity of 9-methyl base-containing sphingolipids was not known. Glycosylceramide containing the 9-methyl-4, 8-sphingadienine base, such as A. oryzae and G. frondosa glycosylceramide, had different effects in decreasing volatile esters, including ethyl caprylate and ethyl 9-decenoate, compared with those produced by soy glycosylceramide containing a 4, 8-sphingadienine base (Tables 1 and 2), suggesting that 9-methyl base in the sphingoid base has a strong inhibitory effects on these volatile flavors. However, there are several phenomena that cannot be explained only by the increase in membrane fluidity, such as the increase in glycerol and decrease in leucine. Sphingolipids are signal molecules that stimulate protein phosphatase 2A (Dobrowsky et al., 1993) and form rafts in the lipid bilayer (Pralle et al., 2000); therefore, glycosylceramide might act via these mechanisms. It will be intriguing to investigate the effect of A. oryzae glycosylceramide on yeasts, which will form the target of our next study.

The behavior, fate, and effects of koji glycosylceramide during fermentation are now starting to be elucidated. Koji contains abundant (0.15–0.25 mg/ml) glycosylceramides (Sawada et al., 2015; Sakamoto et al., 2017). Therefore, sake yeast is exposed to a high concentration of glycosylceramide (40 μg/ml) during fermentation. Koji glycosylceramide is bound to the surface of brewery yeasts, and remains there until the end of fermentation (Sawada et al., 2015). In addition, koji glycosylceramide alters the brewing profiles of sake yeast. These results shed new light on the interaction of koji glycosylceramide and brewery yeasts.

The amount of koji, or the extent of propagation of A. oryzae on the surface of koji (haze), has been empirically considered as an important criterion to govern yeast fermentation by brewing technicians in the sake industry. It has been proposed that mycelia of A. oryzae increase the complex taste of sake, but the precise mechanism has remained unknown. However, together with the data obtained in this study, it can be hypothesized that the amount of lipids, such as glycosylceramide, unsaturated fatty acids, and sterols, contained in koji, is the key to control yeast fermentation.

Sake yeast cells incubated with glycosylceramide showed effects on pyruvate metabolism, the TCA cycle, starch and sucrose metabolism, and glycerolipid metabolism. Lactosylceramide and ceramide cause dysfunction of the mitochondria (Novgorodov et al., 2016; Law et al., 2018); therefore, it might have stimulated the mitochondrial function of sake yeast. Mitochondrial activity has a significant effect on the fermentation profile of sake yeast (Shiroma et al., 2014; Kitagaki & Takagi, 2014; Motomura, Horie & Kitagaki, 2012; Sawada & Kitagaki, 2016); therefore, the upregulation of mitochondrial function in sake yeast might be responsible for the altered fermentation profile. The mechanism of the effect of glycosylceramide on yeast mitochondria requires further research.

Conclusion

In conclusion, we have determined the altered fermentation characteristics of sake yeast in response to glycosylceramide, which will enable interpretation of the effect of koji on the fermentation characteristics of yeast.

Supplemental Information

Supplemental Information 1 Fig. S1. Purification of glycosylceramide from mycelia of A. oryzae.

Click here for additional data file.

Supplemental Information 2 Raw data of flavor and metabolic data.

Data of all flavor and metabolic profiles of sake yeast or media fermented with sake yeast.

Click here for additional data file.

The authors acknowledge Masafumi Kadowaki, Yuki Fujimaru, Marina Ueda, and Eriko Nakahata for their technical assistance.

Additional Information and Declarations

Competing Interests

Author Contributions

Data Availability

The authors declare that they have no competing interests.

Jannatul Ferdouse performed the experiments, analyzed the data, contributed reagents/materials/analysis tools, prepared figures and/or tables, authored or reviewed drafts of the paper, approved the final draft.

Yuki Yamamoto performed the experiments, analyzed the data, prepared figures and/or tables, authored or reviewed drafts of the paper, approved the final draft.

Seiga Taguchi performed the experiments, analyzed the data, contributed reagents/materials/analysis tools, prepared figures and/or tables, authored or reviewed drafts of the paper, approved the final draft.

Yumiko Yoshizaki performed the experiments, contributed reagents/materials/analysis tools, authored or reviewed drafts of the paper, approved the final draft.

Kazunori Takamine performed the experiments, contributed reagents/materials/analysis tools, authored or reviewed drafts of the paper, approved the final draft.

Hiroshi Kitagaki conceived and designed the experiments, analyzed the data, prepared figures and/or tables, authored or reviewed drafts of the paper, approved the final draft.

The following information was supplied regarding data availability:

The flavor and metabolic profile data are available as Supplemental Files.

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
