# Peer review of "Glycosylceramide modifies the flavor and metabolic characteristics of sake yeast"

_PeerJ, doi:10.7717/peerj.4768_

## Round 0.1 · original submission · Major Revisions

Please, address all the changes suggested by the reviewers before your manuscript resubmission, especially those that refer to the description of the methodology.

Reviewer 1 ·

Basic reporting

See attachment

Experimental design

See attachment

Validity of the findings

See attachment

Additional comments

See attachment

Annotated reviews are not available for download in order to protect the identity of reviewers who chose to remain anonymous.

Reviewer 2 ·

Basic reporting

no comment

Experimental design

no comment

Validity of the findings

no comment

Additional comments

This manuscript describes the effects of glycosylceramide on the fermentation of sake yeast. On the whole, findings are new, and it describes sufficient experimental details.

Questions (on the whole)
1. Could you explain why the addition level of glycosylceramides was 40 μg/ml? It should be added to the manuscript. The effects of higher level of glycosylceramides (more than 40 μg/ml) should be added. (The effects of glycosylceramide are dose-dependent?) It helps to consider how level of glycosylceramide is adequate on the flavor and taste of sake.

2. There is no data about the fate of added glycosylceramide in all experiments.

3. In Figure 1, author should explain why the values of “black bars” were different between A and B. For example, acetic acid was 408 % and ap. 120 % in A and B, respectively.

Reviewer 3 ·

Basic reporting

The manuscript by Ferdouse et al reported their study on the effect of Glycosylceramide on metabolic profiles of sake yeast. The English is readable. However, there are major issues with the overall presentation, delivery and methods described. Please see my detailed report

Experimental design

The experimental design is not well described (see my detailed report below). I strongly suggested using a chart

Validity of the findings

It is hard to comment on this as methods used are not very suitable

Additional comments

Except the introduction section, and remaining sections are poorly written. Here are some of my specific comments

1) L82- 83 "Glycosylceramide was purchased from Funakoshi or purified from the mycelia of A. oryzae." Then L84 - L126 almost 5 paragraphs on preparation of Glycosyleceramide from A. oryzae. This is too much focus on this part;

2) A lot of other description is very mechanical, without transitions or explain how it is relevant to the current study

3) Study design is very unclear and not well organized (see #5), I would suggest to use a chart from sample preparation steps to analysis

4) Statistical analysis L190-193 "The differences among the averages of three or more groups were evaluated using analysis of variance (ANOVA), followed by Bonferroni or Dunnett's post hoc multiple-comparison test."
=> What are the 3 or 4 groups? What type of ANOVA? Bonferroni is for multiple testing, post-doc analysis is not! In particular, Dunnett's test accounts for multiple group pairwise comparisons across a single dependent variable, not across the features/metabolites. These two methods are not equivalent

5) L199-200, "To obtain information on the specificity of the effect of chemical structures of glycosylceramide, glycosylceramides from Aspergillus oryzae, soy and Grifola frondosa were used"

=> This is a surprising. It is not even mentioned in the experimental design, but suddenly appear in the results!

6) In the result description L214-L219, it is rather repetitive, without any new information beyond the figure itself

7) Fig1 & Fig2. It seems all two groups (should be t-tests and FDR, rather than Dunnett). I am not sure why Fig 1 and Fig 2 are treated differently in terms of image and statistical analysis. In addition, for many compounds, it is better to use a heatmap rather than many bar plots

8) Fig 3. There is only one compound!

9) Fig 4. "These normalized values were used as independent variables in principal component analysis (PCA) (p < 0.05)."
=> I am confused about PCA and p values here, I could not find other places describe this. Importantly PCA does not give p values!
=> Pathway analysis on 22 detected metabolites for all intracellular and extracellular
The enriched pathways are identical. This is not surprising as these are probably most abundant compounds due to the environment. Their differences are more interesting within the context. i.e. to perform pathway analysis using compounds that are significantly different, instead of using all detected compounds

---

## Round 0.2 · Minor Revisions

There is one reviewer who suggests some remaining changes to your article in order to be accepted. Please, address the suggestions and resubmit the modified version.

Reviewer 1 ·

Basic reporting

See the confidential note to the editor

Experimental design

See the confidential note to the editor

Validity of the findings

See the confidential note to the editor

Additional comments

OK Accepted

Reviewer 3 ·

Basic reporting

The heatmaps (Fig 1 and Fig 2) are uninformative and unprofessional, mostly because the values are not properly normalized to facilitate visualizations of patterns and differences in the compound concentration. Why don't use MetaboAnalyst to generate the heatmaps?

Experimental design

The experimental design chart should be more professionally done and as Fig 1 to help understand the manuscript

Validity of the findings

No comments

Additional comments

Since the current Figures are not informative and there are not so many metabolites, it is more appropriate to have a table listing the concentrations (and p values) of significant metabolites as done in many metabolomics publications (not as supp. material)

---

## Round 0.3 · accepted · Accept

Your article is now Accepted. If you need to make any final copyedit changes please wait until the author´s proof have been made available to you.

#